# Courtship suppression in *Drosophila melanogaster*: The role of mating failure

**Anna A. Goncharova**[1], **Natalia G. Besedina**[1], **Julia V. Bragina**[1], **Larisa V. Danilenkova**[1], **Elena A. Kamysheva**[1], **Sergei A. Fedotov**[1,2,3]*

1 Laboratory of Comparative Behavioral Genetics, Pavlov Institute of Physiology, Russian Academy of Sciences, St. Petersburg, Russia, 2 Laboratory of Toxinology and Molecular Systematics, L.A. Orbeli Institute of Physiology, National Academy of Sciences of the Republic of Armenia, Yerevan, Armenia, 3 Laboratory of Amyloid Biology, Saint Petersburg University, St. Petersburg, Russia

* serg900@yandex.ru

**Data Availability Statement:** All relevant data are within the paper and its Supporting Information files.

## Abstract

*Drosophila melanogaster* is a popular model organism in the study of memory due to a wide arsenal of methods used to analyze neuronal activity. The most commonly used tests in research of behavioral plasticity are shock avoidance associated with chemosensory cues and courtship suppression after mating failure. Many authors emphasize the value of courtship suppression as a model of behavior most appropriate to natural conditions. However, researchers often investigate courtship suppression using immobilized and decapitated females as targets of courtship by males, which makes the data obtained from such flies less valuable. In our study, we evaluate courtship suppression towards immature mobile non-receptive females after training with mated or immature females combined with an aversive stimulus (quinine). We have shown that the previously described mechanisms of courtship suppression, as a result of the association of the courtship object with the repellent, as well as due to increased sensitivity to the anti-aphrodisiac cVA after mating failure, are not confirmed when immature mobile females are used. We discuss the reasons for the discrepancies between our results and literature data, define the conditions to be met in the courtship suppression test if the aim is to analyze the natural forms of behavioral plasticity, and present data on the test modifications to approximate conditions to natural ones.

## Introduction

The courtship behavior of *Drosophila melanogaster* males towards females is an innate behavior that includes a series of actions after detecting and approaching a female. The male taps the female with a foreleg to sample her pheromones, starts to vibrate with one wing (courtship song) and pursuit her, afterwards licks the genitals and attempts to copulate [1, 2]. Courtship provides a male with auditory, mechanosensory, visual, and chemosensory signals from the target fly to assess its suitability (species, sex) and receptiveness [3, 4]. In response to courtship, the receptive female reduces locomotor speed and opens the vaginal plates promoting male copulation attempts [5, 6]. Mated and immature females are non-receptive and exhibit behaviors that prevent copulation including running away, flicking their wings, and other behaviors

**Funding:** This study was supported by the State Program 47 GP "Scientific and Technological Development of the Russian Federation "(2019-2030), theme 0134-2019-0004 and St. Petersburg State University (project #94031363) (SAF). The funders had no role in study design, data collection and analysis, decision to publish, or preparation of the manuscript.

**Competing interests:** The authors have declared that no competing interests exist.

[7, 8]. Courtship activity is measured as the percentage of time spent in courtship out of the total observation time in a testing chamber (courtship index, CI) [9]. In addition to courtship, males can engage in preening, running, or resting [10]. The intensity of courtship towards virgin mature (receptive) and immature (non-receptive) females was reported to remain consistently high for at least an hour or until copulation [9, 11]. At the same time, the intensity of male courtship towards a fertilized (non-receptive) female gradually decreases over 30–60 minutes and copulation attempts fail [9]. 1–3 hours after isolation from a mated female males show courtship suppression when tested with another mated female for 10 minutes, compared to males without experience of prior mating failure experience [12, 13]. This short-term courtship suppression is considered a form of conditioning and is used to study the molecular mechanisms of behavioral plasticity [14, 15]. Protocols and facilities have been developed for mass testing of courtship suppression in flies [16]. Courtship conditioning is also used to test long-term memory lasting several days, in which males are kept with a female for 5–7 hours to perform intermittent multiple courtship attempts [17, 18]. If viewed as conditioning the long-term suppression of courtship in *D. melanogaster* males is similar to conditioned taste aversion [19] and the passive learning models in rodents [20]. In taste aversion, an animal acquires an aversion to the sweet taste of food that was paired with gastrointestinal malaise, and, in passive learning, the animal learns to avoid safe spaces (dark section or bottom area in the chamber) after shocks in them. It has been determined that taste aversion is mostly caused by a specific taste paired with illness [21] and passive avoidance in rats is the conditioned preference for the freezing reaction in an environment associated with electric shocks [22]. While many factors associated with the long-term suppression of courtship in *Drosophila* males have been revealed to date [23, 24], the specific conditioned stimuli for this form of learning have not been identified.

The mechanisms of short-term courtship suppression have been studied in more detail. It has been shown that short-term courtship suppression is caused by increased avoidance of cis-vaccenyl acetate (cVA), which is found on the body of mated females [13]. cVA is contained in the ejaculatory bulb of males [25] and is transferred to the reproductive organs and cuticle of females during copulation [26, 27]. Enhanced avoidance of cVA in courtship tests with mated females (testers) does not depend on the presence of this anti-aphrodisiac during training on a non-receptive female (trainer). Unsuccessful courtship of both mated and immature females has been reported to cause courtship suppression towards a mated female [28]. Changes in courtship behavior are determined by the activity of aSP13 dopaminergic neurons that modulate synaptic connections between Kenyon cells and mushroom body (MB) output neurons M6 in MB gamma lobe [15]. However, the mechanism of aSP13 activation in response to mating failure has not been determined.

One possible trigger for dopaminergic activation may be changes in hormone level in response to unsuccessful courtship attempts. Knockdown of juvenile hormone receptors in dopaminergic neurons has been shown to reduce courtship intensity [29]. In adult *D. melanogaster* males, the level of juvenile hormone positively correlates with the level of 20-hydroxyecdysone (20E) [30], the active metabolite of ecdysone, which is also involved in courtship suppression. Ishimoto et al. [31] revealed that 20E levels increase after courtship conditioning and ecdysone signaling mutants exhibit defects in memory of mate failure. The intensity of courtship is also dependent on the level of ecdysis-triggering hormone (ETH), which provides post-copulation courtship inhibition [32]. Knockdown of ETH receptors in the corpus allatum, where the juvenile hormone is produced, as well as in sensory neurons OR67D and GR32A, which detect the male antiaphrodisiacs cVA and 7-tricosene (7T), respectively, abolishes courtship suppression after mating [32]. Therefore, courtship suppression induced by hormones depends on both the control centers in the fly brain and receptors in the sensory

organs (antennae and labella). In addition, silencing ETH receptors in local interneurons of the antennal lobes also leads to disinhibition of courtship [33].

Since mating failure with a non-receptive female also causes hormonal changes, one can expect changes in sensitivity not only to cVA [13], but also to other antiaphrodisiacs, particularly to 7T, which is detected by GR32A receptors [34]. Moreover, other ligands that bind to GR32A should also increase their effects on fly behavior after unsuccessful courtship. In addition to 7T, GR32A receptors detect several other substances that the flies avoid, including quinine [35, 36]. Previously, Ackerman & Siegel [11] showed that male courtship towards an unreceptive immature female decreased within an hour when quinine was present in the observation chamber. The authors suggested that an association was formed between quinine and aphrodisiacs, leading to reduced courtship stimulation by aphrodisiacs. In their experiment, immature females were used as trainers and paired with quinine in a chamber, while mature females immobilized with ether were used as testers. Courtship suppression was observed in a delayed test without quinine in the chamber, conducted 5–120 minutes after training. The change in courtship targets raises questions about the influence of the specific pheromonal composition of mature females and the smell of ether on the results. In addition, it remains unknown whether courtship suppression would be observed if immature females were used as testers. On the other hand, if the decline in CI during one-hour training with immature females paired with quinine in the work of Ackerman & Siegel [11] is due to non-associative increased sensitivity to quinine, then courtship suppression can be demonstrated in a test with immature females paired with quinine after mating failure with the same type of female without quinine in the chamber.

To answer these questions, we evaluated the suppression of courtship towards non-receptive mobile immature females when quinine was added to the observation chamber. We found that mating failure in the absence of quinine did not increase the suppression of courtship behavior by quinine. To explain the inconsistency of results with our suggestion we compared courtship suppression in tests with mated females after mating failure with immature or mature mated females. Reduction in CI was observed only after training with mated females. Thus, mating failure alone is not sufficient to cause short-term courtship suppression. Furthermore, contrary to the conclusions of Ackerman & Siegel [11] we did not observe decreased CI in the test without quinine after mating failure in the presence of quinine. We discuss the need to revise views on the mechanism of courtship suppression in conditions close to the wildlife and propose a modified courtship suppression test protocol that is more consistent with the behavior of flies in the natural environment. It was revealed that males reared with females according to the protocol demonstrated long-term and short-term behavioral plasticity towards mated females similar to naïve males.

## Materials and methods

### Experimental flies

The wild-type strain Canton-S (CS) of *D. melanogaster* was obtained from the Bloomington Drosophila Stock Center. Flies were maintained on standard yeast-semolina-raisin-sugar medium in plastic vials (95 mm in height and 25 mm in diameter) at 25˚C and under a 12-hour light/dark cycle.

### Experimental conditions

For courtship testing, all flies were collected as virgins without anesthesia by aspiration during 3 hours after eclosion and reared under standard conditions. Experimental males were kept individually for 5 days before the courtship assay unless otherwise specified. Immature females

were used within 14–20 hours after eclosion and kept in a group of 20 flies. Mature females for male training and males for female fertilization were kept in same-sex groups of 20–25 flies per vial. A day before the experiment, 4-day-old females were combined with males of the same age in one vial and left to mate for 18–22 hours.

### Courtship assay

To assess courtship behavior a male and a mobile female were placed by aspiration into a transparent perspex experimental chamber (15 mm in diameter, 5 mm in height) lined with filter paper. An ethogram of male courtship was observed for 300 seconds by identifying and recording the onset of courtship behaviors (orientation, pursuing, vibration coupled with rest or running, tapping, licking, copulative attempts) and non-courtship behaviors (running, preening, and rest). This allowed for the calculation of the courtship index (CI, the percentage of observation time occupied by all courtship elements) and the percentage of time occupied by each behavioral element. Ethogram data were decoded and analyzed using the "Drosophila Courtship Lite 1.4" software developed by Nikolai Kamyshev. All behavioral assays were carried out at least twice on different days by at least two operators at 25˚C between 10 a.m. and 4 p.m.

### Courtship suppression

To assess courtship suppression males were trained with immature or mated mobile females for an hour in the chamber. Courtship in the first and last 5 minutes of one-hour training was recorded. Males that copulated during training were discarded. Sham control males were kept individually in the chamber for an hour instead of training. After training, males were returned to vials for a 30-minute rest and then tested with another non-receptive female (immature or mated) for 5 minutes in a clean chamber. Flies were transferred using shaking and aspiration. To analyze the effects of quinine on courtship suppression 2.0 mg of quinine sulfate (Sigma-Aldrich, 6119-70-6) was brushed onto a 0.5 x 4.2 cm strip of filter paper (Whatman Grade 1) placed around the edge of the chamber.

### Statistics

The comparisons were performed using a randomization test [37]. The null hypothesis was rejected If the *p*-value was less than 0.01. The 95% confidence intervals were calculated using bootstrapping [38] in "Drosophila Courtship Lite 1.4" (10,000 iterations). Spearman rank-order correlation coefficient (Spearman's correlation) was calculated using a *t*-test in IBM SPSS Statistics 20 (Armonk, USA) software.

## Results

### Courtship suppression by quinine is equal in naive and trained males

Courtship suppression was examined in four groups of 5-day-old males. In the first two groups, quinine was added to the chamber during training, and testing was carried out after 30 minutes in one group with quinine in the chamber and in another group without quinine. In the other two groups, testing was similarly carried out with quinine in one group and without in the other, but quinine was not added to the chamber during training.

During the one-hour training with immature females, the male courtship decreased by 32% (Fig 1A). However, in the test conducted 30 minutes after training, the CI returned to the initial value observed in naive males (Fig 1C). If quinine was added to the chamber, the CI in the first 5 minutes of training was almost twice as low compared to males in the chamber without

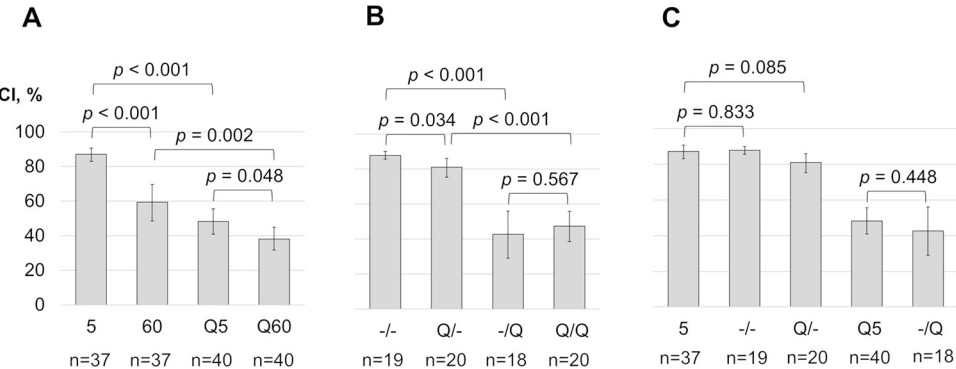

**Fig 1. Courtship indices (CI) of naive and trained males towards immature females.** (A) The CIs are shown within the first 5 minutes (5) and the last 5 minutes (60) of one-hour training, conducted in either a clean chamber or a quinine chamber (5Q and 60Q). (B) The CIs are shown in the courtship suppression test conducted in a clean chamber after training in either a clean (-/-) or quinine chamber (Q/-), as well as in the test where quinine was added to the chamber after training in a clean (-/Q) or quinine chamber (Q/Q). (C) Comparisons of the CIs are provided in clean chambers between the first 5 minutes of training (5) and the tests conducted after training in a clean (-/-) or quinine chamber (Q/-), as well as in quinine chambers between the first 5 minutes of training (Q5) and the test conducted after training without quinine (-/Q). Mean values and 95% confidence intervals are shown. The *p*-values (two-tailed randomization test) are indicated above the respective columns in the comparison, and the number of flies in each group is shown below.

quinine (Fig 1A). At the same time, training in a chamber with quinine did not lead to a significant decrease in courtship time (Fig 1A). These results contradict the dynamics of CI observed with immature females reported by Ackerman & Siegel [11], indicating the need for further study to clarify the methodological and other factors causing this discrepancy. Unfortunately, we did not find other studies that evaluated courtship dynamics with immature mobile females.

We also did not find a significant difference in courtship levels towards immature females in tests conducted after one-hour training in a clean or quinine chamber (Fig 1B). Our data do not support the conclusions of Ackerman & Siegel [11] regarding the formation of an association between female aphrodisiacs and quinine, resulting in courtship suppression. The discrepancies are likely due to differences in courtship objects used in the tests (mature and immature females) and experimental conditions. The courtship suppression observed with mature females after training in a chamber with quinine in the study of Ackerman & Siegel [11] may be influenced by female immobilization or the smell of ether (anesthesia). The use of mature females instead of immature ones in the courtship test seems to be an unlikely reason for the discrepancy between our results and Ackerman & Siegel's [11] data since the authors also observed courtship suppression after training two males in a chamber with quinine and fly carcass coating with an extract of mature females. Therefore, altering the trainer and chemosensory cues did not affect the suppression of courtship towards an immobilized female. An important difference with Ackerman & Siegel's [11] experiment is the different volume of the chamber. In Ackerman & Siegel's [11] work, the chamber had a volume of 0.4 cm$^3$, which is 2.2 times smaller than in the present study (0.9 cm$^3$). The association of quinine with cues of a fly may occur in a more limited space. It is also worth noting that in the experiments of Ackerman & Siegel [11], females were collected under light ether anesthesia 0–10 hours after eclosion while in our experiments all flies were collected without anesthesia by aspiration during 3 hours after eclosion.

We could not confirm our hypothesis regarding the effect of mating failure on courtship suppression values in the test with quinine (Fig 1C). The CI in the test with quinine in the

chamber was 43% after training with immature females, compared to 48% for males with no experience of mating failure (Fig 1C). Keleman et al. [13] demonstrated a non-specific increase in sensitivity to the anti-aphrodisiac cVA after mating failure. To train males, they used pseudomated mature females that behave like mated non-receptive females (remaining virgin) due to signal peptide transgene expression in the nervous system [39]. Therefore, we investigated whether mating failure with immature females would also lead to a decrease in CI in the test with a mated female. As a positive control, mated females were used for training.

## Mating failure with immature females does not suppress courtship in the test with mated females

Unlike training with mated females, training for one hour with immature females did not result in courtship suppression in the test with a mated female (Fig 2A). This finding indicates that mating failure alone is not sufficient to increase sensitivity to cVA on mated females as previously reported by Keleman et al. [13]. We speculate that courtship suppression after training with pseudomated females is likely object-specific. The absence of courtship suppression after mating failure with immature females may explain the negative result in revealing the suggested increase in sensitivity to quinine. Similar to cVA, this effect may only be found in pseudomated females.

We compared the courtship behavior of males with immature and mated females during the first 5 minutes of training. Males trained with immature females were more likely to follow immature females with wing vibration or without vibration (following), less likely to run regardless of the movement of the female, and almost did not attempt copulation (Fig 2B). We suggested that the effectiveness of training depended on the magnitude of parameters that differed between courtship towards immature and mated females. Since these parameters have a high variance in the group with mating females, we estimated Spearman's correlation between the parameters and CIs in the group. The analysis results showed no correlation, leaving the question of courtship suppression factors open. The specificity in rejecting of immature females in response to male courtship should also be considered, particularly the absence of ovipositor extrusion and a higher frequency of fluttering [7].

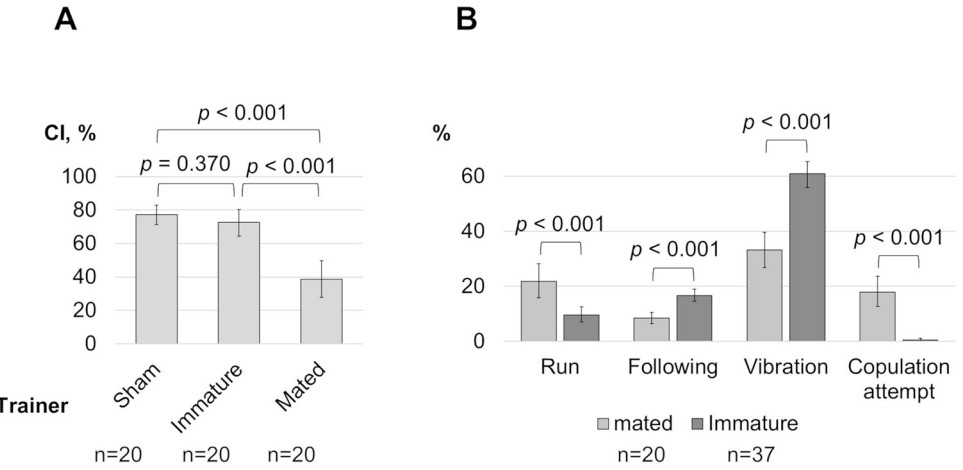

**Fig 2. Comparison of courtship suppression and courtship behaviors during training with immature and mated females.** (A) The CIs of males in tests with mated females are shown after training with immature and mated females. The comparison group consists of males placed in a chamber without a female during the training stage (sham). (B) Percentages of behaviors observed within the first 5 minutes during training with immature and mated females are displayed. See legend in Fig 1.

Thus, the study of courtship suppression using more natural conditions does not confirm findings obtained under conditions that are far from natural (e.g., immobilization of courtship objects, ectopic expression of behavioral factors) and significantly alter male behavior [40]. Since the main advantage of courtship suppression over other memory tests is its naturalness, our results emphasize the need for a more rigorous approach to methodological aspects when studying memory in courtship behavior. In addition to selecting the appropriate courtship object, it is crucial to conduct testing under conditions closer to natural ones. In particular, large observation chambers with the addition of yeast drops can be used, and the courtship behavior of several males towards several females can also be analyzed [7, 28]. However, these solutions require optimization, as they make experiments more expensive and pose challenges for automated analysis. Moreover, optimizing the conditions for fly maintenance before experiments is also necessary. For example, keeping males in a group significantly reduces courtship towards mated females [10]. Most studies isolate males for five days before the courtship suppression test, while females are kept in a group without males. Both conditions differ from the natural conditions of reproduction in mixed groups on a nutrient substrate, which is the case in both natural populations and laboratory stocks. Furthermore, the analysis of the evolutionary role of courtship suppression indicates that this form of learning is associated with the slowly recovering receptivity of mated females and is important for adaptation to high-density populations where females may mate a second time [41].

## Males kept in a group with females learn to suppress courtship like naive males

To approximate fly maintenance for courtship suppression tests to natural conditions, we examined the possibility of using experienced males reared for three days in a mixed group with females. Courtship behavior testing was conducted immediately and with a delay after isolating the males from the group. Mated and virgin mature females were used. Males kept in a mixed group showed reduced courtship towards both mated and virgin mobile females compared to males of the same age kept alone (Fig 3A). The reduced courtship of experienced males towards virgin females was observed for three hours (Fig 3C) and returned to the values in naive males after 24-hour isolation (Fig 3B). At the same time, reduced CI in experienced males towards mated females persisted at least for 24 hours. This indicates the retention of long-term memory about the non-receptivity of mated females rather than the sexual satiety since the courtship level towards virgin females at that time is high. The nature of courtship suppression with virgin females in the first hours after isolation from the mixed group requires further investigation. It is noteworthy that in addition to a reduced CI, experienced males analyzed immediately after isolation also displayed a lower percentage of copulations with virgin females and a longer courtship time before copulation (Fig 3D). The reduced percentage of copulations may be a consequence of reproductive behavior satiety resulting from previous multiple copulations within the group, while the extended courtship time before copulation can reflect the experience of males in courting non-receptive females within the group.

Subsequently, we conducted a courtship suppression test 24 hours after isolating males from the mixed group. One-hour training with mated females resulted in courtship suppression towards mated females but not towards virgin females (Fig 3E). Therefore, our findings indicate that males reared in a mixed group can be used to study both long-term and short-term behavioral plasticity towards mated females under conditions that are closer to natural ones.

## Discussion

Our results do not support the existence of associative forms of plasticity in courtship behavior towards mobile immature flies. Previous studies by Ackerman & Siegel [11] and other authors

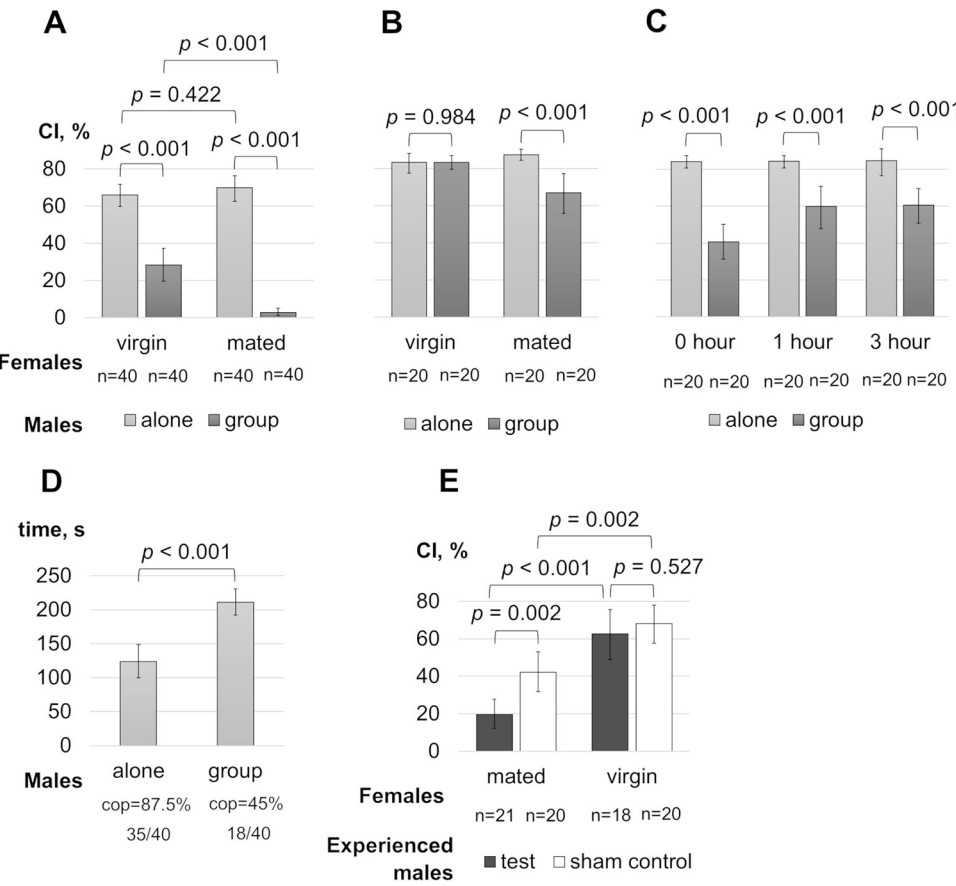

**Fig 3. Courtship of males after being reared in a mixed group with females.** CIs in males kept alone or isolated from the mixed group are shown in immediate (A) and 24-hour delayed (B) courtship assays conducted with virgin and mated females. (C) CIs towards virgin females by males kept alone or isolated from the mixed group at different time intervals (0, 1, and 3 hours) before the assay are shown. (D) The time of courtship towards virgin females before copulation in the immediate assay are shown and the proportion and number of males copulated during 300 seconds of observation are provided under the columns. (E) CIs towards virgin and mated females by experienced males trained with mated females 24 hours after isolation from the mixed group are shown. The comparison group consists of experienced males placed in a chamber without a female during the training stage (sham control). See legends in Figs 1 and 2.

[42, 43] aimed to confirm the hypothesis that courtship suppression was a result of the association between aphrodisiacs and antiaphrodisiacs on the body of mated females [9]. However, these studies used immobilized females to test conditioned avoidance. The females were either decapitated or anesthetized with ether, even if they were non-receptive. Immobilizing the females negated the main advantage of courtship suppression. As a model for the study of learning, courtship was considered a natural form of behavior compared to other tests for memory such as shock-associated odor conditioning [44]. Whether suppression of courtship towards immobilized (e.g., dead) females is a natural form of behavior remains an open question. The authors noted the great variation among trials as the reason for the rejection of mobile receptive females [9, 45]. More recent studies have shown that mating failure does not suppress courtship towards mobile receptive females [13, 28], although opposite results have also been reported [46]. The reasons for these discrepancies remain unclear, but it is important to note that courtship conditioning is highly dependent on the testing conditions. For example, suppression of courtship towards immobilized receptive females is absent when training

with an immature female is conducted in the light, but can be observed when training is conducted in the dark [42]. Similarly, we were unable to confirm courtship suppression towards mated females after training with immature females as previously reported by Dukas [28] who conducted tests in large vials with two males and two females simultaneously.

Ejima et al. [47, 48] investigated courtship suppression in the dark using immobilized immature, virgin, and mated females as trainers and testers in various combinations. Based on the results of these tests, they proposed the formation of an association between mating failure and several undefined conditioned stimuli. Two ways of learning were described: conditioning to a trainer-specific pheromonal profile and generalized cVA-related conditioning (for review, see [49]). In the first way, courtship suppression was observed in tests with immobilized immature or virgin females but only if the training was conducted with the same type of females. The second way is characterized by the suppression of courtship towards all three types of females after training with a mated female. However, attempts to identify the pheromonal profile mediating courtship conditioning have been unsuccessful. Siwicki et al. [43] found a correlation between conditioning with mated trainers and the levels of 9-pentacosene. However, in tests with females lacking cuticular hydrocarbons, adding 9-pentacosene to the chamber was not sufficient to induce courtship suppression in males after mating failure. Therefore, the mechanism of courtship suppression using immobilized females remains undetermined, partly because this form of behavioral plasticity is not natural and involves multiple adaptation mechanisms.

It should be noted that considering mating failure as an aversive stimulus in the work of Ejima et al. [47] does not meet the criteria of associative learning. An association is evaluated based on the reaction to a conditioned stimulus, which is combined in time with the reaction in training and previously did not trigger this response or triggered it extremely weakly [50]. Therefore, considering mating failure as a stimulus is incorrect, as failure itself is not a sensory signal triggering a reaction. The nonspecific sensitization to anti-aphrodisiac and hormonal shifts observed by Keleman et al. [13] and Ishimoto et al. [31] after mating failure indicate a change in the functional state of the organism, in which the same stimuli elicit different innate reactions (avoidance) that are not the result of association. Similarly, inserting a nipple into an infant's mouth triggers a sucking reflex if the infant is hungry, but after being satiated, the infant exhibits nipple spitting and other innate reactions to avoid feeding [51]. Interestingly, short-term suppression of courtship towards mobile immature males is also a form of non-associative learning resulting from habituation to aphrodisiacs on the cuticle of immature individuals [52]. Thus, the mechanisms of short-term courtship suppression with mobile flies such as pseudomated females and immature males have been described. On the other hand, the explanation for the courtship suppression with immobilized flies remains at the level of hypotheses. In Fig 4 we have illustrated published data on the presence or absence of courtship suppression in various combinations of trainers and testers, as well as on the mechanisms of suppression.

It is important that the associative suppression of courtship can, in principle, be developed, for example, in response to external cues from non-receptive females, such as eye color [55]. At the same time, this form of conditioning is achieved by approaching the conditions of training and testing as close as possible to natural ones. Males are placed for training in a group with receptive and non-receptive mobile females with different external characteristics and are tested in a day with two receptive mobile females that are different in these characteristics.

It is still unknown whether long-term courtship suppression in a standard assay using mated females [17] results from non-associative or associative avoidance. In the passive avoidance model in rats, an association is formed between environmental cues and electric shocks, which inhibit relocation to a safe place and increase freezing response [22]. However, the

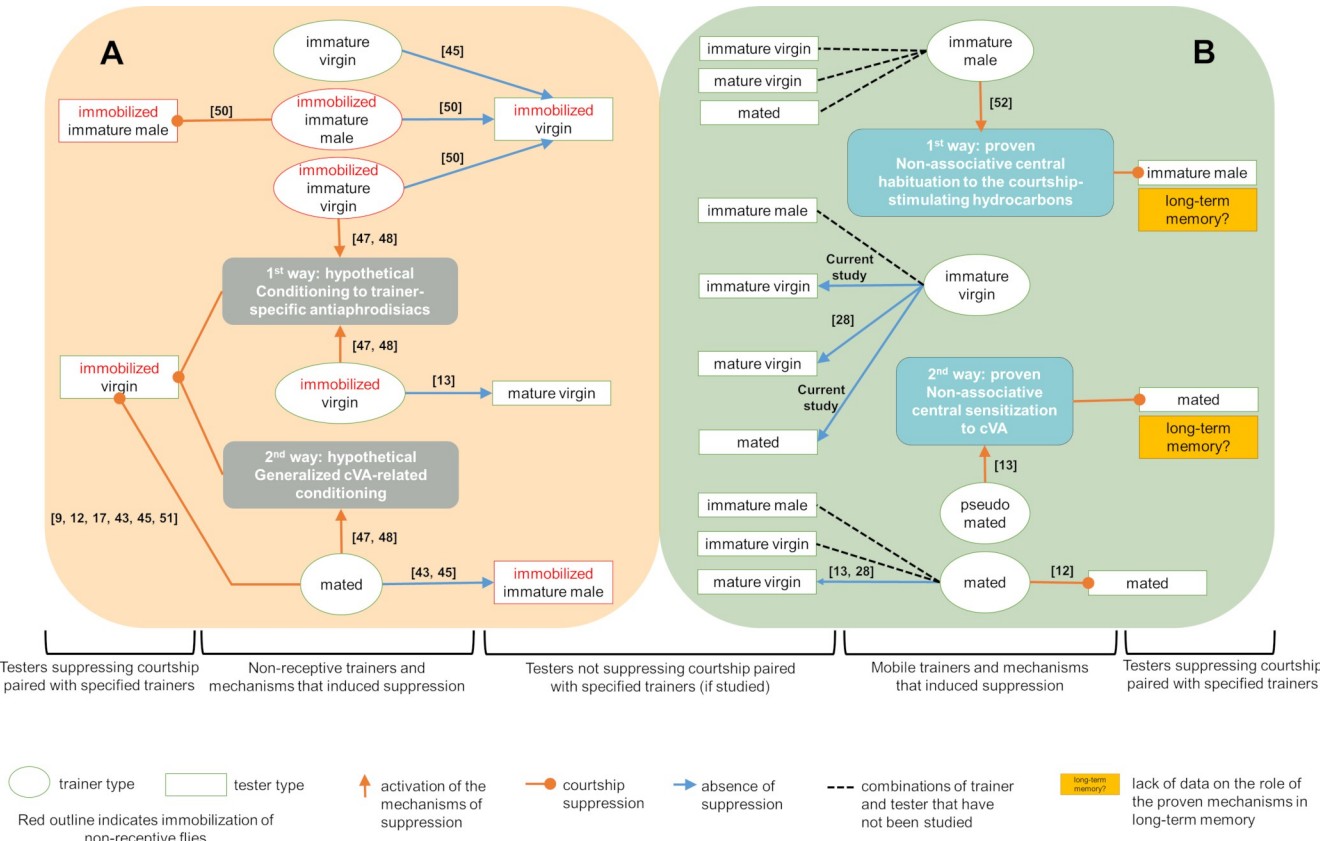

**Fig 4. Studies of short-term courtship suppression and its mechanisms in various combinations of trainer and tester flies (female if not otherwise indicated).** (A) Combinations described in the literature [9, 12, 13, 17, 43, 45, 47, 48, 53, 54] are illustrated where the trainer and/or tester flies are immobilized. (B) Combinations described in the literature [12, 13, 28, 52] and all other possible combinations are illustrated where both the trainer and tester flies are mobile. A review of the objects, conditions, and conclusions of the mentioned studies can be found in a previous review [49].

specific cues that are crucial for long-term courtship suppression have not yet been identified (Fig 4B). It is important to note, that in experiments on long-term courtship suppression, researchers often use immobilized flies [23, 24]. Considering that there is no data on the courtship towards immobilized females in nature, this approach may be ineffective for studying memory mechanisms, as was the case with short-term memory. For example, inconsistencies in courtship assay results were observed in studies involving juvenile hormone-deficient males, which differed in mobility and maturity. Wijesekera et al. [56], using immature females, showed that males with reduced levels of juvenile hormone acid O-methyl transferase have a reduced CI, whereas Lee et al. [57] found no decrease in CI in males tested with decapitated mature virgin females.

## Conclusion

In the present study, using immature mobile females, we examined the mechanisms of courtship suppression described in the literature. Male courtship activity was tested after a training session with the addition of a repellent and after mating failure. Our data do not support the formation of an association between the aversive stimulus (quinine) and the aphrodisiacs of immature females, resulting in courtship suppression. Similarly, mating failure with immature females did not lead to courtship suppression with the same type of females. Our results

highlight the dependence of behavioral plasticity mechanisms on various factors, including the mobility and maturity of females. Researchers studying courtship suppression as a natural model of memory should strive to use testing conditions that closely resemble the natural breeding environment of flies.

## Supporting information

**S1 Appendix. Experimental data.** All data used to generate the results for this study. (XLSX)

## Acknowledgments

The authors would like to acknowledge Nikolai Kamyshev for his valuable comments and the Resource Center "Chromas" at St. Petersburg State University for their technical support.

## Author Contributions

**Conceptualization:** Sergei A. Fedotov.

**Data curation:** Anna A. Goncharova, Natalia G. Besedina, Julia V. Bragina, Sergei A. Fedotov.

**Formal analysis:** Anna A. Goncharova, Julia V. Bragina, Sergei A. Fedotov.

**Investigation:** Anna A. Goncharova, Natalia G. Besedina, Larisa V. Danilenkova, Elena A. Kamysheva, Sergei A. Fedotov.

**Methodology:** Anna A. Goncharova, Natalia G. Besedina, Julia V. Bragina, Sergei A. Fedotov.

**Project administration:** Julia V. Bragina.

**Resources:** Anna A. Goncharova, Julia V. Bragina.

**Supervision:** Sergei A. Fedotov.

**Validation:** Anna A. Goncharova, Sergei A. Fedotov.

**Visualization:** Sergei A. Fedotov.

**Writing – original draft:** Sergei A. Fedotov.

**Writing – review & editing:** Anna A. Goncharova, Julia V. Bragina, Sergei A. Fedotov.

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
