## [Decision Letter · Decision Letter 0]

29 Jun 2023

PONE-D-23-16725Courtship suppression in Drosophila melanogaster: the role of mating failurePLOS ONE

Dear Dr. Fedotov,

Thank you for submitting your manuscript to PLOS ONE. After careful consideration, we feel that it has merit but does not fully meet PLOS ONE’s publication criteria as it currently stands. Therefore, we invite you to submit a revised version of the manuscript that addresses the points raised during the review process.

We look forward to receiving your revised manuscript.

Kind regards,

Efthimios M. C. Skoulakis, PhD

Academic Editor

PLOS ONE

Journal Requirements:

Additional Editor Comments:

Clearly both reviewers find the manuscript very interesting and impactful and I concur. However, both reviewers raise questions and pose requests aiming to further clarify the insights gained and interpretations from the work presented and to further increase its impact in the field. Please address these comments and requests carefully and thoroughly.

Reviewers' comments:

Reviewer's Responses to Questions

**Comments to the Author**

1. Is the manuscript technically sound, and do the data support the conclusions?

Reviewer #1: Yes

Reviewer #2: Yes

2. Has the statistical analysis been performed appropriately and rigorously? 

Reviewer #1: I Don't Know

Reviewer #2: No

3. Have the authors made all data underlying the findings in their manuscript fully available?

Reviewer #1: Yes

Reviewer #2: Yes

4. Is the manuscript presented in an intelligible fashion and written in standard English?

Reviewer #1: Yes

Reviewer #2: Yes

5. Review Comments to the Author

Reviewer #1: In their paper entitled “Courtship suppression in Drosophila melanogaster: The role of mating failure“, Goncharova et al examine how “courtship failure”, ie a male courting unsuccessfully during an initial “training phase”, affects/suppresses his subsequent courtship. They used two main approaches to study these effects:

(1) They examined a male that was exposed to a mobile immature, non-receptive female, with the aversive chemical quinine present at the same time. They then tested the male’s subsequent courtship to another immature virgin female. In the presence of a live immature female, where copulation will not occur (“courtship failure”) male courtship decreased 32% in an hour. When quinine was present, courtship was significantly decreased, but did not further decline over an hour.

When males that had experienced courtship failure were tested after training they showed a high courtship index again. Males that had been trained in the presence of quinine showed normal courtship as well when tested after training. The presence of quinine during testing reduced courtship to the reduced levels observed in training when quinine was present.

Thus, the authors did not find a reduction of courtship in subsequent tests when males experienced courtship failure in the presence of quinine in their paradigm. These findings suggest that in this paradigm no association of the courtship object/failure with the repellent occurred.

These findings differ from findings by other authors that had found an association of the courtship object and the repellent that resulted in courtship repression in a subsequent test. The authors discuss the differences in the training situation that likely contribute to the observed differences. They suggest that the mobility of the training object made a major difference in the outcome, since in experiments described by others the training object was immobilized.

Questions / requests: The authors should discuss in more detail other difference between the two approaches. For example, mature females or animals coated with cuticular extracts were used as trainers in previous assays, and subsequent tests were with immature and mature females. Were quinine concentration and application the same? How about the timing? A careful and detailed comparison may allow additional important insight into mechanisms that underlie courtship suppression, an important aspect of courtship plasticity.

(2) In a next experiment, the authors paired males in a group with immature females for three days and then assessed their courtship towards an immature or a mated female immediately afterwards, or 24 hours later. Immediate courtship was suppressed in both cases. However, courtship of immature females was normal after 3 hours, whereas courtship towards mated females took 24 hours to regain normal levels. The authors suggest that this difference reflects the formation of different kind of associations and could be used to test different forms of memory.

Questions/requests: Notably in this paradigm, the male will gain experience with immature, mature and mated females, since he will eventually encounter all three in the vial as the females mature and will become perceptive. Could the authors speculate how this affects the outcome?

How does satiety influence the observations or the experimental setup for this assay? How can it be separated from memory functions?

(3) In figure 4, the authors provide a graph with an extensive review of previous results and their own findings. This is a valuable resource. It also very complex and not easy to read for non-experts.

Request: I would suggest that the authors try to add more clarity. For example, by clearly designating the trainers (what animals participate in the training set up) and likewise in the testing. What exactly is being tested? What was concluded?

General comments and suggestions:

Overall, the experiments are well described and carefully performed. The authors’ conclusions are supported by the data.

I believe that a careful look at how these results differ from previously described ones will likely provide insight into possible mechanisms of courtship suppression under various conditions. This will be helpful in understanding this plasticity in natural as well as experimental settings, and as a tool to study molecular mechanisms underlying it.

I would suggest doing so in a less combative way than it is presently done in the paper; this will enhance the discussion on pros and cons of different experimental approaches.

I doubt that others using immobilized females in their assays would argue that this is a natural conditions, or that it is used for this reason, but that it leads to robust associations that are useful for the examination of conditioning and memory pathways. The reason for using this natural behavior is often because it is a strong innate behavior and thus more robust than other forms of associative learning (odor/shock paradigms). As is often the case, the best approach may differ depending on the questions asked. If the question focuses on wildtype situation, the approach might be different than when genetic pathways and mutants are interrogated.

Reviewer #2: This interesting paper deals with discrepancies found in the bibliography after using the Courtship Conditioning learning paradigm. Such differences might be due to technical changes in the protocol itself. The authors compare several conditions in order to prove the different hypothesis. Their main findings are that there is no association between courtship object and an anti-aphrodisiac, that courtship failure is not decisive and that experiences males show stronger and more robust responses. Specially interesting is the discussion, very insightful. I believe that the paper deserves to be published but some minor changes need to be addressed.

1.- From my point of view, the title does not reflect the findings of the paper. Although it is interesting to determine that mating failure is not essential in this learning paradigm, I found fascinating that mated males show a better response than virgin males- I agree with authors that this should be more similar to natural conditions, but it should be noted that training with mobile immature female is unlikely to occur in nature. However, no information about this result is found in the abstract or in the conclusions.I wonder if the title might be changed reflecting also this last, interesting result. Also, the abstract need to be modified reflecting the main conclusions of the paper, including the differences between experienced and virgin males.

2.- The last paragraph of introduction is quite complex, it is not easy to understand. Needs some clarification and should include the results from fig 3. Also I recommend to use a native english specialist, to amend some grammar mistakes

3.- In fig 1 and fig 2A the p-value is used instead the A-NOVA, which seems more appropriate given that three or more datasets are compared. Also, please also show in fig 1a the CI of 60 min. Is it different?

4.- Fig 4 needs to be modified. Indicate the legend for each signal (Ellipses – trainer typee) in the figure and not in the figure legend. Also, it is not clear what means the figure: explanation for the two hypothesis? summarizing data? Please clarify.

5.- When discussing the immature virgen female results, I suggest to read and comment the paper by Verzijden et al, 2015 (Current Zoology), where they describe a novel learning paradigm using immature and mature virgin females. The success in mating is associated with a particular external appearance of females. Also it is not mentioned the extensive review from Kramer´s lab (Raun et al, 2021, J of Neurogen). I believe that discussion might be benefitted from its reading. Also, part of the discussion is in the result part (lines 224 to 239)

6- Conclusion needs to add the main finding that experienced males exhibit a more robust response to CC, which is an important result.

6. PLOS authors have the option to publish the peer review history of their article (what does this mean?). If published, this will include your full peer review and any attached files.

Reviewer #1: No

Reviewer #2: No

---

## [Author Response · Author response to Decision Letter 0]

19 Jul 2023

Dear Dr. Efthimios Skoulakis,

We are grateful to reviewers for critical reading of the manuscript and their comments. Accordingly, we have revised the manuscript essentially. The changes in the manuscript are highlighted. Responses to reviewers' comments are submitted.

We ensured that our manuscript meets PLOS ONE's style requirements, including those for file naming (according to The PLOS ONE style templates https://journals.plos.org/plosone/s/file?id=wjVg/PLOSOne_formatting_sample_main_body.pdf)

Upon re-submitting our revised manuscript, we uploaded our study’s minimal underlying data set as Supporting Information file “S1_Appendix.xlsx”.

Sincerely,

Dr. Sergei Fedotov

Responses to comments of Reviewer #1

Dear Reviewer, thank you for your critical reading of the manuscript and your suggestions for its improvement. Below are our responses.

In their paper entitled “Courtship suppression in Drosophila melanogaster: The role of mating failure“, Goncharova et al examine how “courtship failure”, ie a male courting unsuccessfully during an initial “training phase”, affects/suppresses his subsequent courtship. They used two main approaches to study these effects:

(1) They examined a male that was exposed to a mobile immature, non-receptive female, with the aversive chemical quinine present at the same time. They then tested the male’s subsequent courtship to another immature virgin female. In the presence of a live immature female, where copulation will not occur (“courtship failure”) male courtship decreased 32% in an hour. When quinine was present, courtship was significantly decreased, but did not further decline over an hour.

When males that had experienced courtship failure were tested after training they showed a high courtship index again. Males that had been trained in the presence of quinine showed normal courtship as well when tested after training. The presence of quinine during testing reduced courtship to the reduced levels observed in training when quinine was present.

Thus, the authors did not find a reduction of courtship in subsequent tests when males experienced courtship failure in the presence of quinine in their paradigm. These findings suggest that in this paradigm no association of the courtship object/failure with the repellent occurred.

These findings differ from findings by other authors that had found an association of the courtship object and the repellent that resulted in courtship repression in a subsequent test. The authors discuss the differences in the training situation that likely contribute to the observed differences. They suggest that the mobility of the training object made a major difference in the outcome, since in experiments described by others the training object was immobilized.

Questions / requests: The authors should discuss in more detail other difference between the two approaches. For example, mature females or animals coated with cuticular extracts were used as trainers in previous assays, and subsequent tests were with immature and mature females. Were quinine concentration and application the same? How about the timing? A careful and detailed comparison may allow additional important insight into mechanisms that underlie courtship suppression, an important aspect of courtship plasticity.

Answer: In our work, we tried to reproduce Ackerman & Siegel’s experiments as accurately as possible, including using the same quinine concentration and the same delayed time during which the suppression was observed. Other difference between the two approaches is added and highlighted (revised manuscript: lines 195-205).

(2) In a next experiment, the authors paired males in a group with immature females for three days and then assessed their courtship towards an immature or a mated female immediately afterwards, or 24 hours later. Immediate courtship was suppressed in both cases. However, courtship of immature females was normal after 3 hours, whereas courtship towards mated females took 24 hours to regain normal levels. The authors suggest that this difference reflects the formation of different kind of associations and could be used to test different forms of memory.

Questions/requests: Notably in this paradigm, the male will gain experience with immature, mature and mated females, since he will eventually encounter all three in the vial as the females mature and will become perceptive. Could the authors speculate how this affects the outcome?

How does satiety influence the observations or the experimental setup for this assay? How can it be separated from memory functions?

Answer: Dear reviewer, sorry, but we must clarify, first, that we used mature virgin (not immature) and mated females in the male courtship test after rearing in a group (see line 261). Second, mating female courtship did not recover to normal levels after 24 hours (see line 265). Influence of mixed rearing and separation of satiety from memory functions are considered and highlighted (revised manuscript: line 265-267). 

(3) In figure 4, the authors provide a graph with an extensive review of previous results and their own findings. This is a valuable resource. It also very complex and not easy to read for non-experts.

Request: I would suggest that the authors try to add more clarity. For example, by clearly designating the trainers (what animals participate in the training set up) and likewise in the testing. What exactly is being tested? What was concluded?

Answer: Designation of trainers, testers and changes in courtship level are added in Fig. 4 (see revised Fig4). The purpose of the figure is to show the presence or absence of courtship suppression in various combinations of trainers and testers and the mechanisms of suppression (according to literature data). Unfortunately, it is not possible to indicate objects, conditions and the conclusions within the figure, but we referred to a work of Ruan et al. (2021) reviewing most of the studies mentioned.

General comments and suggestions:

Overall, the experiments are well described and carefully performed. The authors’ conclusions are supported by the data.

I believe that a careful look at how these results differ from previously described ones will likely provide insight into possible mechanisms of courtship suppression under various conditions. This will be helpful in understanding this plasticity in natural as well as experimental settings, and as a tool to study molecular mechanisms underlying it.

I would suggest doing so in a less combative way than it is presently done in the paper; this will enhance the discussion on pros and cons of different experimental approaches.

I doubt that others using immobilized females in their assays would argue that this is a natural conditions, or that it is used for this reason, but that it leads to robust associations that are useful for the examination of conditioning and memory pathways. The reason for using this natural behavior is often because it is a strong innate behavior and thus more robust than other forms of associative learning (odor/shock paradigms). As is often the case, the best approach may differ depending on the questions asked. If the question focuses on wildtype situation, the approach might be different than when genetic pathways and mutants are interrogated.

Answer: Dear reviewer, thank you for this valuable comment.

Responses to comments of Reviewer #2

Dear Reviewer, thank you for your critical reading of the manuscript and your suggestions for its improvement. Below are our responses.

This interesting paper deals with discrepancies found in the bibliography after using the Courtship Conditioning learning paradigm. Such differences might be due to technical changes in the protocol itself. The authors compare several conditions in order to prove the different hypothesis. Their main findings are that there is no association between courtship object and an anti-aphrodisiac, that courtship failure is not decisive and that experiences males show stronger and more robust responses. Specially interesting is the discussion, very insightful. I believe that the paper deserves to be published but some minor changes need to be addressed.

1.- From my point of view, the title does not reflect the findings of the paper. Although it is interesting to determine that mating failure is not essential in this learning paradigm, I found fascinating that mated males show a better response than virgin males- I agree with authors that this should be more similar to natural conditions, but it should be noted that training with mobile immature female is unlikely to occur in nature. However, no information about this result is found in the abstract or in the conclusions. I wonder if the title might be changed reflecting also this last, interesting result. Also, the abstract need to be modified reflecting the main conclusions of the paper, including the differences between experienced and virgin males.

Answer: Dear reviewer, sorry, but we did not demonstrate that mated males show a better response (courtship index? courtship suppression?) than virgin males. Figure 3A shows decreased CI level of experienced males towards virgin and mated females in comparison with naïve males. 3B shows decreased CI level of experienced males towards mated females after 24-hour isolation (in comparison with naïve males). Figure 3C shows decreased CI level of experienced males towards virgin females immediately and after 1-3-hour isolation (in comparison with naïve males). 3D shows increased time of courtship for experienced males copulated in assay with virgin female (naive males achieved copulations faster). At last, 3E shows courtship suppression only in experienced males towards mated females and absence of courtship suppression towards virgin females after one-hour training with mated females. We have made slight changes in the Fig.3 to present the results more clearly (indications for type of females and males are added, see Fig3_revised).

2.- The last paragraph of introduction is quite complex, it is not easy to understand. Needs some clarification and should include the results from fig 3. Also I recommend to use a native english specialist, to amend some grammar mistakes

Answer: The clarifications and the result from fig 3 have been added and highlighted in the last paragraph of Introduction (revised manuscript: line 116-124). Grammar corrections have been made throughout the revised manuscript.

3.- In fig 1 and fig 2A the p-value is used instead the A-NOVA, which seems more appropriate given that three or more datasets are compared. Also, please also show in fig 1a the CI of 60 min. Is it different?

Answer: ANOVA assumes that each sample was drawn from a normally distributed population. Our datasets do not follow normal distribution. Two statistical tests of normality – Kolmogorov-Smirnov and Shapiro-Wilk – generate significance value less than the alpha value 0.01 (IBM SPSS Statistics 20). The randomization test, which we used to compare mean values, does not require the normality of distributions or equality of variances. It is based on Monte-Carlo approach and calculates the probability of null-hypothesis directly by repeated resampling the real data sets (we used 10000 iterations). We could compare the samples by nonparametric methods. But they take into account only ranks and, thus, are less sensitive than randomization test.

Dear reviewer, sorry, but fig 1a already demonstrates the CI of 60 min towards immature females when they were tested in a clean chamber and in a chamber with quinine. The CIs in these two experiments are different (p value is indicated above the respective columns).

4.- Fig 4 needs to be modified. Indicate the legend for each signal (Ellipses – trainer typee) in the figure and not in the figure legend. Also, it is not clear what means the figure: explanation for the two hypothesis? summarizing data? Please clarify.

Answer: Designation of trainers, testers and changes in courtship level are added in Fig. 4 (see Fig4_revised). The purpose of the figure is to show the presence or absence of courtship suppression in various combinations of trainers and testers and the mechanisms of suppression (according to literature data). Clarifications are added in the revised manuscript (lines 337-339).

5.- When discussing the immature virgen female results, I suggest to read and comment the paper by Verzijden et al, 2015 (Current Zoology), where they describe a novel learning paradigm using immature and mature virgin females. The success in mating is associated with a particular external appearance of females. Also it is not mentioned the extensive review from Kramer´s lab (Raun et al, 2021, J of Neurogen). I believe that discussion might be benefitted from its reading. Also, part of the discussion is in the result part (lines 224 to 239)

Answer: Many thanks for the suggestion to read Verzijden et al, 2015 (Current Zoology). This study demonstrates a proven courtship conditioning mechanism and points to the need to test males in natural conditions (competitive assay). Verzijden et al, 2015 is commented in lines 346-351.

Review from Kramer´s lab (Raun et al, 2021, J of Neurogen) is mentioned in lines 313, 345.

Dear reviewer, we would like to leave the discussion from lines 224 to 239 (239-255 in revised version) in the results because it provides a rationale for the test modifications that we propose and analyze in the next section of the results.

6- Conclusion needs to add the main finding that experienced males exhibit a more robust response to CC, which is an important result.

Answer: Dear reviewer, sorry, but we did not demonstrate that experienced males show a more robust response to CC (courtship conditioning?) than virgin males. Please, see comments to the first question.

---

## [Decision Letter · Decision Letter 1]

1 Aug 2023

Courtship suppression in Drosophila melanogaster: the role of mating failure

PONE-D-23-16725R1

Dear Dr. Fedotov,

We’re pleased to inform you that your manuscript has been judged scientifically suitable for publication and will be formally accepted for publication once it meets all outstanding technical requirements.

Kind regards,

Efthimios M. C. Skoulakis, PhD

Academic Editor

PLOS ONE

Additional Editor Comments (optional):

Reviewers' comments:

Reviewer's Responses to Questions

**Comments to the Author**

1. If the authors have adequately addressed your comments raised in a previous round of review and you feel that this manuscript is now acceptable for publication, you may indicate that here to bypass the “Comments to the Author” section, enter your conflict of interest statement in the “Confidential to Editor” section, and submit your "Accept" recommendation.

Reviewer #1: All comments have been addressed

Reviewer #2: (No Response)

2. Is the manuscript technically sound, and do the data support the conclusions?

Reviewer #1: Yes

Reviewer #2: Yes

3. Has the statistical analysis been performed appropriately and rigorously? 

Reviewer #1: Yes

Reviewer #2: Yes

4. Have the authors made all data underlying the findings in their manuscript fully available?

Reviewer #1: Yes

Reviewer #2: Yes

5. Is the manuscript presented in an intelligible fashion and written in standard English?

Reviewer #1: Yes

Reviewer #2: Yes

6. Review Comments to the Author

Reviewer #1: In my view the authors have addressed concerns and questions. Figure 4 is greatly improved. It is still complex, but so are the summarized data

Reviewer #2: The authors addressed all my comments. I thank them for clarifying my misunderstanding and making the text clearer. The paper is acceptable for publication.

7. PLOS authors have the option to publish the peer review history of their article (what does this mean?). If published, this will include your full peer review and any attached files.

Reviewer #1: No

Reviewer #2: No

---

## [Editor Report · Acceptance letter]

3 Aug 2023

PONE-D-23-16725R1 

Courtship suppression in *Drosophila melanogaster*: the role of mating failure 

Dear Dr. Fedotov:

I'm pleased to inform you that your manuscript has been deemed suitable for publication in PLOS ONE. Congratulations! Your manuscript is now with our production department. 

Kind regards, 

on behalf of

Dr. Efthimios M. C. Skoulakis 

Academic Editor

PLOS ONE